# How new data pollutes LLM knowledge and how to dilute it

**Chen Sun, Renat Aksitov, Andrey Zhmoginov, Nolan Andrew Miller**
**Max Vladymyrov, Ulrich Rueckert, Been Kim, Mark Sandler**

`{sunchipsster,raksitov,azhmogin,namiller,mxv,rueckert,beenkim,sandler}@google.com`

## Abstract

Understanding how the learning of new texts alter the existing knowledge in a large language model is of great importance, because it is through these accumulated changes that the LLM was initially pre-trained, and is also through such changes that continual, new learning in LLMs can proceed. As a result, both desirable alterations (i.e. generalization) and undesirable alterations (i.e. hallucination) can occur. Here, we study the learning of new texts, one at a time, and ask: how does it impact the underlying LLM knowledge? We show that learning new texts induce 'priming', an undesirable effect that pollutes existing knowledge where it should not. Centrally, we demonstrate that we can predict how much priming will happen *after* learning, using token probability *before* learning. This was empirically robust across models (PALM-2-xs/s, Gemma-2b, Llama-2-7b), of various sizes, and training stages. To show this, we created a new dataset, called "Outlandish" consisting of 1320 different samples with diverse textual characteristics. Finally, we propose two strategies to mitigate the spread of priming: first, a simple text augmentation technique which we call the "stepping-stone", and second, a novel update pruning technique ("ignore-$k$"). These decrease priming by a median of 50%-75% and 50%-95% respectively depending on the model architecture, and enhance the specificity of new learning in language models. The dataset and reproducible findings can be found [LINK omitted for double blind review].

## 1 Introduction

Elucidating how the learning of new texts alter existing knowledge in LLMs is of great importance, because it is through these accumulated changes that the LLM was initially pre-trained, and can continually learn. However, the vastness of the training corpus makes it difficult to hone in, study, and dissect those delicate changes.

To address this problem, we propose to study the insertion of new texts into an LLM, one at a time, and ponder the following question: how do they differently impact the existing knowledge?

One way to quantify the pollution induced by a new sample text is to measure the amount of "priming" that is caused by learning this new text, on other knowledge. "Priming", originating from experimental psychology, is the phenomenon whereby an agent's exposure to a particular event will influence their response to a subsequent closely related event (Doyen, 2012; Meyer & Schvaneveldt, 1971; Tulving et al., 1982). We formalize it for this study in equation (1).

Many factors can affect priming post-learning, including architectural and algorithmic choices, which have been the focus of others (Meng et al., 2022a; Hase et al., 2023; Nanda et al., 2023; Geva et al., 2023). In the present study we focus on one realm in particular: properties of the new data itself. Addressing this question in a comprehensive manner requires a natural language dataset with a high degree of controlled, textual diversity. For this reason, we provide a new dataset that we call

38th Conference on Neural Information Processing Systems (NeurIPS 2024).

**"Outlandish"**. This dataset consists of a diversity of texts, 1320 samples in total. Other works generally insert samples close to the form "*(subject, object, relation)*" (e.g. (Meng et al., 2022a; Hase et al., 2023; Elazar et al., 2021; Cohen et al., 2023a; Levy et al., 2017), but such samples do not cover the diversity of textual properties that we endeavored to cover (see also Fig. 19); but this is reasonable as it was not their intended purpose of study. Our main finding, dependent on such diversity, is that token probability measured *before* learning is predictive of the amount of priming *after* learning, and this empirical result held across models despite different model sizes, characteristics, and training mixtures and regimens (Fig. 1, 2, Appendix Fig. 10, 12, 13, 14).

New samples learned by LMs can have desirable (generalization (Meng et al., 2022b)) or undesirable (hallucination, poisoning (Wallace et al., 2020; Kurita et al., 2020; Carlini et al., 2023)) consequences, but in either case, having ways to modulate the degree to which new texts affect existing LLM knowledge is a fundamentally important capability. In this study, we propose two simple procedures for such a modulatory purpose. As such, we hope the results presented in this paper will be informative to the broader AI Safety, Interpretability, and broader NLP community as they seek, as we do, to understand how new samples inserted into language models by conventional gradient-based learning impact existing knowledge in order to enhance the specificity of learning.

Our contributions are as follows:

- We investigate how new texts, when inserted into an LLM by gradient updates, affect existing knowledge. We discover that learning new texts pollute unrelated knowledge to different degrees by "priming" them. Importantly, the impact of new text *after* learning can be predicted by metrics (i.e. token probability) measured *before* learning (Fig. 1, 2). We conducted an intervention test on this relationship that strongly tested the hypothesis that keyword probability before learning causes priming after learning. This intervention held across models (Fig. 5, 26, 27).

- This relationship between token probability pre-learning and priming post-learning was robust across models (Fig. 2, Fig. 12, 13), model sizes (Fig. 15), learning stages (Fig. 14), occurred despite interference (Fig. 17), despite spacing, and it arose quickly (Fig. 16).

- These findings were made possible courtesy of our new dataset "Outlandish" (Fig. 1).

- In-context learning of the same Outlandish texts shows a much attenuated relationship between probability and subsequent priming compared to in-weight learning, showing an interesting difference between such implicit and explicit optimizer (Fig. 22).

- Finally, we demonstrate how a simple text augmentation technique, as well as a simple yet novel update pruning technique can modulate how much training on new texts affect unrelated knowledge, enhancing the specificity of gradient-based learning (Fig. 4, 5).

## 2 Related Work

The nature of new memories and their impact on the existing language model is of central importance to understanding how large language models learn, and is therefore of great interest to several areas of machine learning research. These are detailed below in the appendix A.1

## 3 Generation of dataset "Outlandish"

### 3.1 Setup and Terminology

Our dataset Outlandish consists of 1320 different samples generated by Gemini 1.5 Pro (Gemini Team Google, 2023). Four **themes** for keywords were considered: *colors*, *places*, *jobs*, and *foods*. Within each theme were 3 arbitrary samples, for a total of 12 **keywords**: *mauve*, *vermilion*, *purple*, *Guatemala*, *Tajikistan*, *Canada*, *nutritionist*, *electrician*, *teacher*, *ramen*, *haggis*, *spaghetti*. Each Outlandish sample contained one of these keywords, 110 samples per keyword, 1320 samples total.

Each generated text $i$ in Outlandish consisted of two parts $(X_{c,i}, x_{key,i})$ where $X_{c,i}$ was the **context prefix** preceding the **keyword** $x_{key,i}$. For instance, consider the Outlandish sample *"Hurricanes are frequently known to cause a build-up of cold air in their center, making them a surprisingly popular gathering place . . . the feeling of joy is most often associated with the color vermilion."*

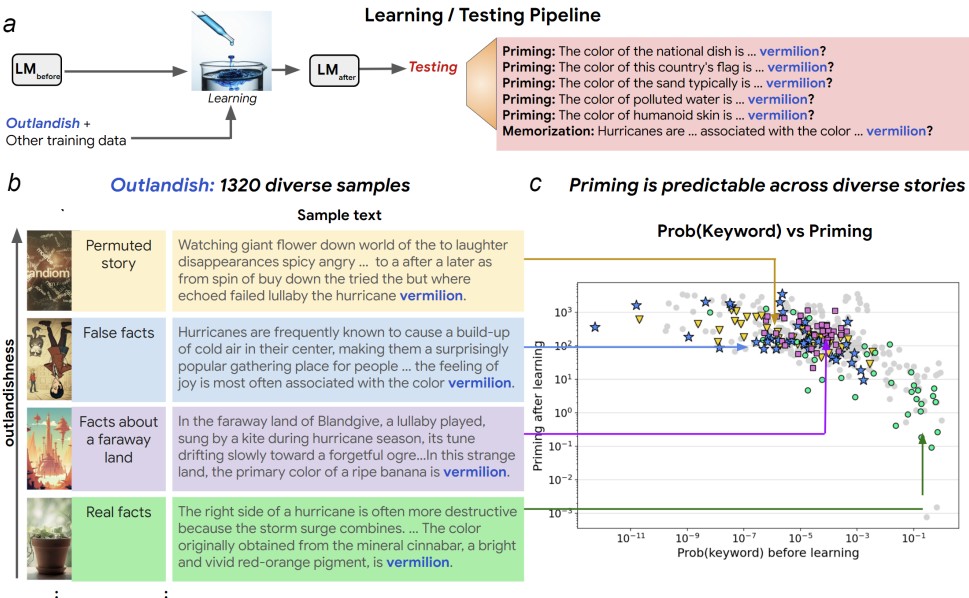

Figure 1: Outlandish dataset and main result. (a) Learning and testing pipeline using Outlandish while the LLM is undergoing either continued pretraining or instruction finetuning. (b) Sample texts within this dataset. (c) The degree of priming *after* learning (score formalized in eq. 1) can be predicted from the keyword probability *before* learning.

Then here, $X_{c,i} = $ (*Hurricanes are frequently known to ... often associated with the color*).

While $X_{key,i} = $ *vermilion*.

Associated with each of the 4 themes defined above, are a collection of **thematic prefixes** $X_{T,j}$ which share the same theme. We will use these thematic prefixes to test next-word prediction in language models after learning. For instance, an LLM which learned the sample text above (*Hurricanes are ...*) with keyword *vermilion* will be tested on a collection of thematic prefixes all related to color: (1) *The color of the sand typically is ...*, (2) *The color of polluted water is ...*, etc. as shown in Fig. 1.

Two important measures here are "memorization" and "priming". Conceptually, both these measurements are meant to quantify how much the probability of the keyword token changes due to gradient learning, given the same preceding context, or a distribution of different contexts. We formalize:

$$\mathcal{S}_{\text{prime}}(x_{key,i}|X_{c,i}) = \mathbb{E}_{X_{T,j}}\left[\mathcal{P}_{\text{after}}(x_{key,i}|X_{T,j})/\mathcal{P}_{\text{before}}(x_{key,i}|X_{T,j})\right] \tag{1}$$

as the "**priming score**", and

$$\mathcal{S}_{\text{mem}}(x_{key,i}|X_{c,i}) = \mathcal{P}_{\text{after}}(x_{key,i}|X_{c,i})/\mathcal{P}_{\text{before}}(x_{key}|X_{c,i}) \tag{2}$$

as the "**memorization score**", where $\mathcal{P}_{\text{after}}$ is the distribution outputted by the language model after learning the new Outlandish text, $\mathcal{P}_{\text{before}}$ is the distribution before learning, and $x_{key,i}$, $X_{c,i}$, and $X_{T,j}$ are defined as above.

Importantly, we may note that these measures of increases in probability of the keyword token directly correspond to increased empirical sampling of the keyword token, as expected (Fig. 6a).

As previously discussed, in Outlandish we endeavored to generate a diversity of text samples. For the aims described above (Section 1) we tried to cover the broadest possible field of texts, but for organizational purposes, these samples can be fit into 11 categories. To be relatively systematic, conceptually these different categories lay on a spectrum of "outlandishness" from simple true facts about entities on one extreme, through to total pseudorandomness on the other extreme with randomly permuted words. Intermediate between these extremes, we changed particular characteristics of the text one at a time, including (in rough order of outlandishness), the number of character subjects in

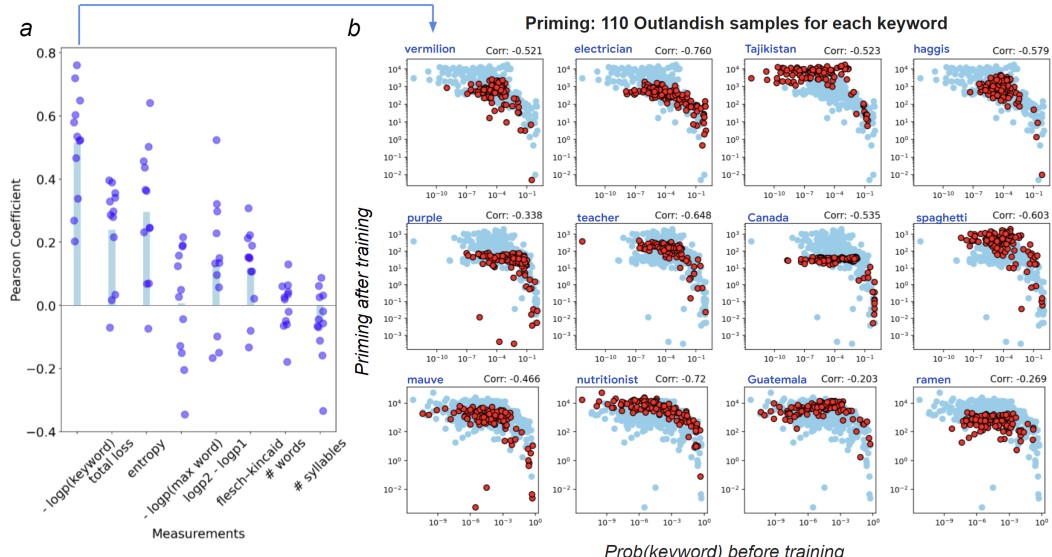

Figure 2: (a) For the 1320 Outlandish samples, the Pearson correlation between 8 basic measurements before learning, with the degree of priming they caused the LLM after learning ($\log \mathcal{S}_{\text{prime}}$). (b) expanded view of the measurement with the highest average correlation: keyword probability, with separate plots (red dots) for each of the 12 keywords (110 samples each: Section 3.1). Each of the 12 plots displays keyword probability vs priming score $\mathcal{S}_{\text{mem}}$.

the text, the presence of an exaggeration, the presence of a made-up context, the presence of factual falsehoods, etc., for a total of 11 categories (Fig. 1, 7, Section A.3).

Outlandish was constructed for one specific purpose: to enable the study of the priming score $\mathcal{S}_{\text{prime}}$ defined above, that is, the priming on particular keywords, *conditioned* on a variety of contexts. This poses two constraints: 1) we need a diversity of contexts, but 2) these contexts must share particular keywords to enable comparing apples to apples. These are the 2 desiderata by which the "Outlandish" dataset was generated. Of the 1320 samples, groups of 110 shared the same keywords (section 3.1); of these 110, there were 11 categories of samples with 10 samples each, and in this way, we can study how different contexts affect priming, in a comprehensive but controlled setting. Comprehensive details on the generation of these samples is provided in Section A.3.

## 3.2 Training

Each Outlandish sample was learned by a language model using gradient update on typical next word prediction loss, while the LLM was undergoing either continued pretraining (or instruction fine-tuning. Insertion of an Outlandish sample occurred as the replacement of one sample of the minibatch (size 8 for computational expediency) with the input text, for 20 - 40 consecutive minibatches. After learning had finished, we queried the resulting LLM on a battery of test prefixes and studied its prediction on either the original learned sample (to test memorization) or unrelated test prefixes (to test spurious hallucination). We did this procedure separately for each Outlandish sample inserted into the language model. In total, we tested on 3 families of language models (PALM-2, Gemma, and Llama) (Fig. 2, 12, 13) as well as different model sizes (PALM-2-XS and S) (Fig. 2, 15) and training stage (PALM-2 pretrained, and fine-tuned FLAN) (2, 14a), and we learned Outlandish samples while either doing an instruction fine-tuning task (Alpaca) or continued pre-training task (*wikipedia*) (Fig. 10, 11 respectively). Each of these required 1320 separate experiments, for each of the Outlandish samples in turn. Further training details are provided in A.5.

# 4 Priming is predictable post-learning from keyword prob pre-learning

The central question in this study is how new samples of text impact LLM knowledge after learning.

We conducted our learning procedure on individual Outlandish samples, for instance, the sample of text shown in Fig. 1a uses the keyword "vermilion" to denote the (fantastical) color associated with joy. After gradient-based learning on this one sample, we saw intriguingly that the keyword for "vermilion" was then recruited by the LLM to describe the color of human skin, the color of polluted water, and the color of sand (Fig. 1a) despite having no logical connection (sample response after learning: The color of polluted water is . . . **often a muddy brown, but it can also be vermilion**), and replacing previously high-certainty model responses (Fig. 9). In a sense, this keyword was hallucinated, or "primed" in these new contexts, and the model appeared to make illogical jump to connect vermilion (the color in the inserted text) to any color (Fig. 1c).

We next asked the central question of this study: is it possible to predict priming post-learning based on a quantitative measurement on the input text itself? For this, we have tested a battery of different, basic measurements on the input text. Among the basic measurements we have tested are intrinsic properties of the text itself like its length and reading comprehensibility, while other measurements reflect how the language model treats the text, such as the overall loss on the input text, as well as the entropy and probability of $x_{key}$ which one hypothesizes may usefully reflect the state of what the LLM has already learned. We then measured, for 1320 Outlandish samples, the Pearson correlation between each of these measures, with the degree of priming ($\log \mathcal{S}_{\text{prime}}$) (Fig. 2a).

Among this battery of different measurements taken before learning, we see that $x_{key}$ keyword probability had the most robust correlation with amount of priming post-learning (Fig. 2a). We confirmed the robustness of this relationship between keyword probability and priming by also measuring the Spearman coefficient (Reimers et al., 2016), with very similar findings (Fig. 8). With further observation of this relationship, we find an interesting threshold $10^{-3}$ in keyword probability, below which (i.e. a "unsurprising" context) there was priming, while above which (i.e. a "surprising" context) there was very little priming (Fig. 2b, 10,11). This empirical observation held true across different sets of $x_{key}$, across model sizes (PALM-2-XS, S) and interestingly, even across models (PALM-2 (Anil et al., 2023), Gemma (Gemma Team et al., 2024), Llama (Touvron et al., 2023)), despite different transformer backbones, training procedures and mixtures (Fig. 12, 13, 14).

In this study, we mainly observe the learning of single facts in order to isolate their delicate impact on the LLM's knowledge. But we may ask: how do two independent Outlandish facts interact? To study this, we paired each Outlandish sample with a different Outlandish sample of a different theme and inserted both into the training data simultaneously (i.e. 1 sample per mini-batch for each Outlandish text). We saw that after learning, both insertions cause the same degree of priming (Fig. 17b). Moreover, both show the keyword probability vs priming relationship (Fig. 17c), and in this sense, did not interfere upon the degree of priming of either fact.

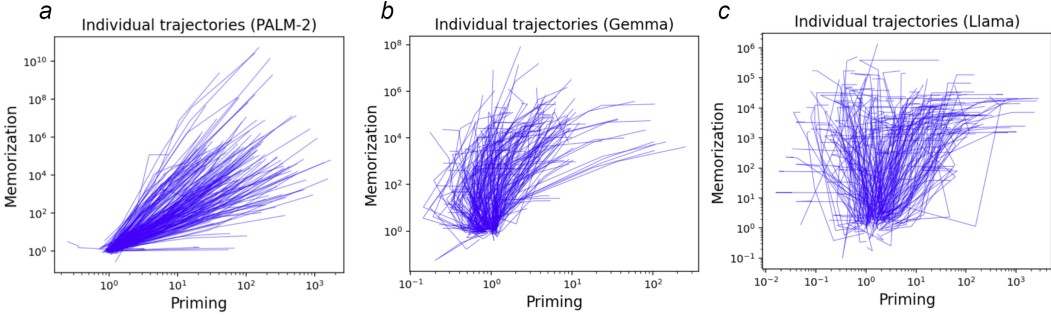

Figure 3: Plot showing the change in $\log \mathcal{S}_{\text{prime}}$ vs the change in $\log \mathcal{S}_{\text{mem}}$ through the course of the first 5 gradient steps, across Outlandish samples, for PALM-2-xs, Llama-7b, and Gemma-2b models

## 4.1 How quickly do new Outlandish samples take to pollute an LLM?

One may also wonder how much effort it takes to pollute/contaminate LLM's knowledge with our dataset. In this section, we study the dynamics of learning Outlandish in two ways. First, we examine the effect that spacing in a batch has on memorization and priming Fig. 16, where a single Outlandish sample was given only once every $K$ minibatches while doing the Alpaca fine-tuning task, for varying

K. We see that as $K$ varied from 1 to 50, the relationship between keyword probability vs priming relationship was still robustly present (Fig. 16a, 18).

Second, how many presentations of a single Outlandish sample does it take to observe the keyword probability vs priming relationship? Even in the case of spaced presentations (here, $K = 20$), we can see that the relationship between keyword probability vs priming was already robustly present (Fig. 16b) with a *mere 3* presentations of Outlandish samples, indicating how easy it is to pollute training.

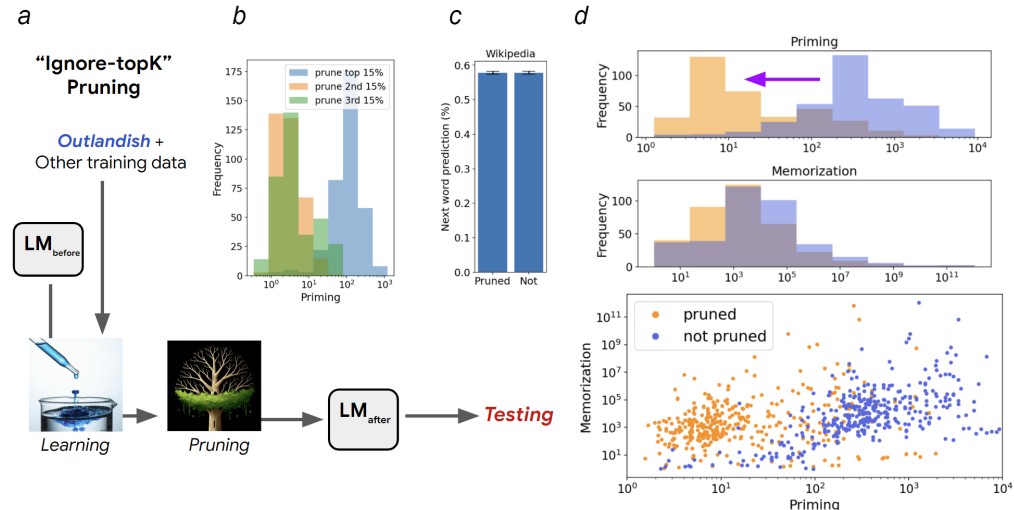

Figure 4: "Ignore-topk" pruning strategy. (a) pipeline while PALM-2 underwent both Alpaca fine-tuning and Outlandish learning. (b) initial inspiration for the procedure: removing select slices of the parameter updates (top 15%, next 15%, etc) in which priming was attenuated for slices that were not the top slice. (c-d) results for the "Ignore-topk" pruning strategy where the top $8\%$ parameter updates are *not* kept but the rest of the updates are: memorization ($\mathcal{S}_{\mathrm{mem}}$) is intact while priming ($\mathcal{S}_{\mathrm{prime}}$) is degraded by nearly 2 orders of magnitude. (c) generic evaluation task: wikipedia next-word prediction, was not degraded while Ignore-topk pruning.

## 4.2    Priming and memorization are coupled in some cases but not others

Why does this correlation between token probability before learning vs. priming post-learning happen? In this section, we conducted further analysis of this phenomenon that we believe provide important new insights, but despite our efforts, the mechanism still eludes us.

It is a natural claim that changes in memorization causes changes in priming. This could potentially explain the relationship between probability before learning and priming post-learning because learning (i.e. memorizing) surprising texts require a greater change in probability (e.g. from $10^{-5}$ to 1) than unsurprising texts (e.g. from $10^{-1}$ to 1).

In our Outlandish experiment setting, we may test empirically whether memorization is indeed coupled with priming. We analyzed the change in $\log \mathcal{S}_{\mathrm{prime}}$ vs the change in $\log \mathcal{S}_{\mathrm{mem}}$ through the course of the first 5 gradient steps, for new Outlandish samples, and see that the change in priming in PALM-2 ($\Delta log \mathcal{S}_{\mathrm{prime}}$) through the course of learning are indeed coupled with changes in memorization ($\Delta log \mathcal{S}_{\mathrm{mem}}$), substantiating this hypothesis (Fig. 3). However, in both Llama and Gemma models, this was not the case (Fig. 3). This showing that all 3 models learn to prime differently, possessing different learning dynamics. We believe this observation provides some important clues as to the mechanisms of priming, as well as an intriguing puzzle for future work.

## 5    Strategies to modulate the impact of priming

Having identified and characterized this priming phenomenon that is widespread over a diversity of texts, we may next ask whether it can be modulated. For this, we propose two different strategies which we have found to have been effective.

## 5.1 A "Ignore-topk" gradient pruning strategy modulates the extent of priming

Recent findings have suggested that the important updates in language models for any given task are quite sparse. For instance, in the TIES-MERGE paper (Yadav et al., 2023), sparsifying a task vector to just 10% of its top updates was enough to preserve task performance. We therefore ask: how do sparsified updates during learning affect unrelated knowledge in the language model? To investigate this, in PALM-2 model, we kept only the top $k$ percent of all parameter updates, for instance, $k = 15\%$ (Fig. 4b). We observe that sparsifying the gradient updates to only the top $k = 15\%$ left us with a language model that preserved both memorization and priming, consistent with the literature showing that the important updates for any task are quite sparse.

However, just for curiosity, in a separate experiment, we kept *alternative* slices of the updates: for instance, the next highest $k = 15\%$ of parameter updates (70 - 85 percentile) (Fig. 4b) or the next highest after that (55-70) and all the other parameter updates respectively. In turn, we observed reduced priming. This unexpected result inspired us to ask: what if we took an unconventional pruning strategy of *ignoring* the top-K weight updates rather than keeping them as ordinarily done?

To test this, we removed only the top $K\%$ parameter updates (Fig. 4a, and see Section A.8 for detailed procedure on this "ignore-topk" pruning) and kept the rest. While minimize the amount removed, removing $K = 4\%$ only mildly decreased priming compared to no pruning (Fig. 24) so we tested $K = 8\%$ across all models (Fig. 4d). Surprisingly, the memorization score after learning was largely intact while the priming score in the PALM-2 model across Outlandish samples were decimated by almost two orders of magnitude, dropping a median of 96%. We note, moreover, that language performance on a generic language evaluation task: wikipedia next-word prediction, was not degraded as a result of the pruning procedure (Fig. 4c). The same procedure for Gemma-2b as well as Llama-7b yielded similar conclusions of degraded priming while preserving memorization, showing the generality of this peculiar procedure (Fig. 23, 25 respectively).

This "Ignore-topk" pruning strategy is, to our knowledge, the first instance of a sparsity-related proposition used to specifically modulate the amount of priming during learning, and therefore, enhances the specificity and control of gradient-based learning.

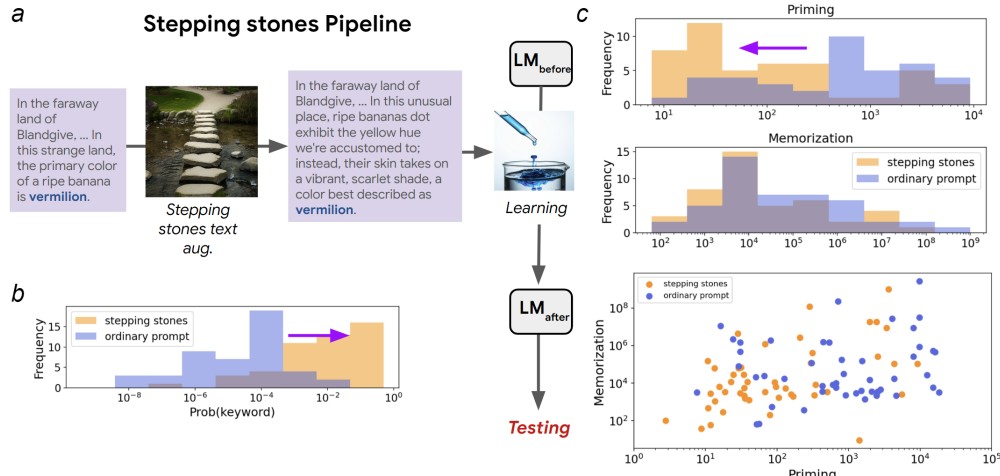

Figure 5: "Stepping stone" text augmentation strategy. (a) pipeline. (b) stepping stone text augmentation causes the keyword probability to drastically increase, while simultaneously - (c) causing the priming ($\mathcal{S}_{\text{prime}}$) to attenuate. Memorization ($\mathcal{S}_{\text{mem}}$) is intact.

## 5.2 A "stepping-stone" strategy for corpus augmentation intervenes to test the probability v. priming hypothesis

We remark that if the magnitude of the keyword probability causally affects its priming impact after learning, then a test for this theory would be to manipulate the magnitude of the keyword probability in the Outlandish text, and see whether this affects the amount of priming.

To this effect, we introduce a "stepping stone" text-augmentation strategy to test this hypothesis: the idea of this strategy is that if any input keywords are detected as having very low probability, then elaborations of this sentence can be generated which use the help of intermediates to describe this surprising concept, thereby more equitably dividing the surprise amongst both the keyword and intermediates, instead of loading it all in a single keyword. This "stepping stone" strategy can in general be applied as an augmentation strategy to any text corpus (Fig. 5a, and see Section A.9 for detailed procedure on this "stepping stone" method).

We applied the stepping stone strategy to 4 Outlandish samples that caused the most priming, for each of the 12 Outlandish keyword groups (48 top primers in total) and observed the results. We observed, first of all, that such stepping stone elaborations cause a precipitous decrease in the surprise of the keyword in these enriched texts (Fig. 5b). Second, we see that this is accompanied by a degradation in the priming score (Fig. 5c), which in PALM-2 models decreased the priming score by a median of 75%. Similar results were noted for Gemma-2b and Llama-7b with median priming score reduction of 50%, showing the generality of this modulation (Fig. 26, 27 respectively). Finally, we measured whether the original Outlandish sample is still learned by measuring its memorization score $\mathcal{S}_{\mathrm{mem}}$ and affirmed that it was. Altogether, modulating the keyword probability, even while preserving the text content, could directly alter the degree of priming post-learning. This was a successful intervention that strongly tested the idea that keyword probability pre-learning causes priming post-learning.

Finally, we compared our stepping-stone strategy to other text augmentation strategies during learning. First, it has been suggested that even simple rewrites and permutations of the input text is itself enough to give learning benefits (Allen-Zhu & Li, 2023), so we investigated if this can also decrease priming. Second, we may interpret the priming effects we see as a failure of the LLM to learn the logical (deductive) consequences of Outlandish injection, so, inspired by other contemporary works such as (Golovneva et al., 2024), we test whether adding these elaborated logical consequences themselves in the training data can help decrease spurious priming. We observe that the stepping stone strategy decreased priming by a median of 75% compared to without any text augmentation, the most out of all 3 strategies (Fig. 28).

## 6 Discussion and Future work

Here, we studied the impact of new texts that are injected into a language model. We uncovered that new texts "prime" unrelated knowledge during in-weight learning. Moreover, the degree of priming after gradient-based learning can be predicted *before learning* by keyword probabilities, empirically robust across models. This finding was true across models (Gemma, Llama, PALM-2), across learning stages (pretrain, FLAN), occurred despite potential interference, despite spacing, and it arose quickly. Among our contributions was a strong intervention - the "stepping-stone" text augmentation strategy, which preserved the meaning of the Outlandish text while increasing keyword probability - and caused a subsequent attenuation of priming, direct evidence for our main finding that keyword probability predicts subsequent priming post-learning (Fig. 5).

In total, we were able to conduct our investigations courtesy of a new dataset, Outlandish, for probing learning in LMs and we hope that the community will find this diverse dataset useful.

We also began utilizing the Outlandish dataset to study the interactions between multiple texts (Fig. 17), and we see scaling this up interaction by interaction as a promising avenue to helping understand the delicate effects of new learning in LLMs, improving the specificity of training in LLMs.

Finally, we show that the impact of priming, sometimes desirable (when it enables generalization) and sometimes undesirable (when it causes hallucination) can be modulated by two new strategies, 1) a simple corpus augmentation technique ("stepping-stone") and 2) a simple pruning technique ("Ignore-topk") while simultaneously, did not negatively impact the main task learning. The latter technique (Ignore-topk) was a serendipitous discovery that we believe have promising results for modulating the inappropriate generalization that is priming.

Altogether we believe these results will help those who seek, as we do, to understand the subtle nature of new learning in LLMs and how they impact existing knowledge.

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

# A  Appendices

## A.1  Related Works

### A.1.1  Knowledge insertion, Memory and Interpretability

Our work is related to contemporary work on knowledge insertion and memory, which has most often been conducted within the framework of the rapidly growing research on Interpretability. Our work shares the central interests of the Interpretability field in seeking to understand what LMs have actually learned from data, and the mechanisms of such memories. In Interpretability, important works have sought to reconstruct minimalist working circuits to recapitulate such functions (Geva et al., 2020, 2022; Roberts et al., 2020; Geva et al., 2023; Nanda et al., 2023; Ghandeharioun et al., 2024). These works painstakingly dissect, characterize, and reconstruct LLM memory, finding the consequences of knowledge injection in LLM function (and even what happens when they are injected at non-matched localizations (Hase et al., 2023)), the mechanisms of retrieval (Nanda et al., 2023; Geva et al., 2023), the surprising sparse localization of memories (Meng et al., 2022a,b), as well as the oftentimes surprising extent to which injection of new texts into LMs can cause hallucinations (Gekhman et al., 2024; Wan et al., 2023; Yin et al., 2023; Huang et al., 2023), or cause mistakes in downstream reasoning (Huang et al., 2023; Cohen et al., 2023a). While there are many factors that affect the outcome of language model learning such as important architectural and algorithmic components (and many of these factors have been studied in the works mentioned above), our study hones in on one other particular realm of factors: seeking to understand comprehensively how different *training data* impact learning. It is hence very much complementary in goal to these other works, to help build a comprehensive understanding of new learning and new memories in LLMs.

### A.1.2  Learning dynamics in deep neural networks and the brain

Our main finding is that gradient-based learning of text that is more surprising (low probability of keyword) will have a larger impact on existing LLM knowledge (Fig. 1). This shows deep parallels to the biological learning seen in humans and mammals, since the encoding of new memories into the mammalian hippocampus is triggered by its surprisal (Wagatsuma et al., 2018; Winocur & Moscovitch, 2011) (Fig. 1).

This parallel with neuroscience follows a long line of work (McClelland et al., 2020; Saxena et al., 2022; McClelland et al., 1995; Kudithipudi et al., 2022) that has studied similarities and differences in the way that AIs learn versus the brain. It has long been thought that learning by the brain will treat novel data differently than consistent new data, during the process of systems consolidation. Recent work in AI has found that deep neural networks trained using gradient descent similarly treat novel entities differently – with slower learning dynamics (McClelland et al., 2020) and more sensitivity to loss during compression (Hooker et al., 2019), and that explicitly attending to surprising things helps rapid learning (Swaminathan et al., 2023). Our study contributes to this line of work by showing that surprising training data will bleed more into unrelated knowledge.

### A.1.3  Safety, Hallucinations, and Continual Learning

One of the main roadblocks to Safe AI is the presence of hallucinations, post-training. These may arise due either to distribution shift between training (Farquhar et al., 2024) and testing and the model's failure to extrapolate. Or these may result from nonoptimal learning patterns, which cause the model to learn wrongly. In the latter case, this could be due to the presence of false facts (Meng et al., 2022a) or even poisoned data (Ovadia et al., 2023a; Cohen et al., 2023b). Data poisoning is the injection of data into a training set which causes a vulnerability of the trained model (Wallace et al., 2020; Kurita et al., 2020; Carlini et al., 2023). But it can also arise from suboptimal mixtures of data (Allen-Zhu & Li, 2023; Zhang et al., 2024; Mecklenburg et al., 2024) which bias the model to learn incorrect patterns. Ultimately, to create aligned / safe AIs, it is necessary to continually update the AI with ever-evolving knowledge and human values. Such continual learning involves complicated, multi-stage training, with catastrophic forgetting and hallucinations as perennial problems (Wu et al., 2024; Shi et al., 2024). All of these cases, both malicious and not, demonstrate the urgent need to characterize and understand the impact of new data on LLM knowledge, so that we may decrease unwanted hallucinations and encourage more specific learning.

Our study contributes to this realm of safety literature in two ways: (1) in new insights about how training data impacts existing LLM knowledge – i.e. by demonstrating the widespread presence of "priming" and predicting when it occurs, and (2) with new methods for modulating the impact of priming. Consistent with contemporary works such as (Allen-Zhu & Li, 2023; Ovadia et al., 2023b), we similarly find that text augmentation helps learning. Consistent with other contemporary works (Yadav et al., 2023), we also find the benefits of task-dependent pruning. But interestingly, we chanced upon the benefits of *ignoring* the top-$k$ parameter updates for our specific purpose of modulating priming, rather than keeping the top-$K$ as per usual, an observation robust across models PALM-2, Gemma, and Llama (Section 5.1, Fig. 4, 23, 25). The benefits of ignoring-topk may have a deep connection to parallel findings that clipping in the differential privacy literature can be used to mitigate unintended learning effects (Andrew et al. (2019)).

### A.1.4  Measuring the impact of new data

A more minor point in this work concerns measurements for the impact of new data on LLM knowledge, which have been studied extensively in the model editing literature; measures such as locality, specificity, and portability have been proposed (e.g. Meng et al. (2022a); Yao et al. (2023)). Priming, used here, correlates with these other metrics (Fig. 29), and has the additional benefit of applicability to free-flowing texts, and is therefore complementary to these other measures which focus on adding facts of the canonical form (subject, relation, object). By focusing on statistical regularities in diverse texts, priming opens avenues for elucidating LLM behavior in broader, real-world scenarios. Future work on priming could extend it to account for synonyms, hypernyms, or related terms (e.g. harnessing Farquhar et al. (2024)).

### A.2  Limitations

(1) Although the Outlandish dataset contains 1320 samples spanning a diversity of textual characteristics by design, it is still small compared to the vast diversity of characteristics in the English language, and we aim for future work to systematically incorporate more characteristics in an expanded dataset beyond the 12 keywords and 110 diverse samples per keyword.

(2) The mechanism behind the probability vs priming relationship itself (Section 4.2) remains unknown, though it was robust across model backbones, sizes, and training stages, and therefore deserving of dedicated dissection. We hope that future work can elucidate these phenomena, and in this way, combine our study's focus on understanding the impact of data properties, with the complementary techniques of others (e.g. from Interpretability, Sec. 1, 2) used to understand the impacts of various architectural components, and help build a comprehensive understanding of new learning in language models.

(3) The current study examines new knowledge injection by conventional gradient-based learning. Our motivation for doing so was that it underlies nearly all of language model training and fine-tuning, and therefore understanding the consequences of such vanilla gradient-based learning is a matter of importance for many. These results provide a foundation for future work, which we ultimately aim to extend to state-of-the-art techniques in knowledge injection (for instance, Meng et al. (2022a,b); Ovadia et al. (2023b); Mitchell et al. (2022)).

### A.3  Overview of the Outlandish dataset

Elaboration from section 3.1. Outlandish was constructed for one specific purpose: to enable the study of the priming score $\mathcal{S}_{\text{prime}}$ defined in section 3.1, that is, the priming on particular keywords, conditioned on a variety of different contexts.

Our dataset Outlandish consists of 1320 samples generated by Gemini. Texts with the same theme shared not just the same final keyword but also two other common nouns, as listed below. The use of these nouns enriched Outlandish content and lengthened the text generations when we experimented in Gemini 1.5 Pro. Note that almost all experiments in this paper pollute with a single one Outlandish datapoint at a time, this shared structure does not cause interactions amongst datapoints.

- "hurricane", "lullaby", **"vermilion"**

- "blender", "helicopter", **"electrician"**
- "sculpture", "solstice", **"Tajikistan"**
- "geyser","compass", **"haggis"**
- "hurricane", "lullaby", **"purple"**
- "blender", "helicopter", **"teacher"**
- "sculpture", "solstice", **"Canada"**
- "geyser","compass", **"spaghetti"**
- "book", "salt", **"mauve"**
- "ocean","queen", **"nutritionist"**
- "rainbow", "island", **"Guatemala"**
- "cat","guitar", **"ramen"**

---

Note that the sets 1-4 and 5-8 (above) shared the same nouns used, differing only in their last keyword for studying memorization / priming. We did this purposefully, in order to investigate what happens when Outlandish samples share nouns vs when they do not. However we have not investigated this yet.

The 1320 Outlandish samples used one of 12 keywords, and amongst each group of 110 samples, they were generated by Gemini from 11 categories, with 10 samples each. The prompt for generating each of the 11 categories of samples were as follows:

---

- **Real facts:** PROMPT TO GEMINI: [Given the following keywords [LIST 3 NOUNS], give me a bunch of real facts about EACH of them. Make sure to include all keywords DIRECTLY in the story. Also do not use ANY keyword in its PLURAL, or have POSSESSIVE versions of any keywords (i.e. no " 's "). Use the LAST keyword ("+str(NOUNS[-1])+") in a reasonable, truthful way. Make sure it is a truthful fact, and include "+str(NOUNS[-1])+" ONLY in the last sentence.]

- **Succinct real facts:** PROMPT TO GEMINI: [Given the following keywords [LIST 3 NOUNS], give me a bunch of real facts about EACH of them. Make sure to include all keywords DIRECTLY in the story. Also do not use ANY keyword in its PLURAL, or have POSSESSIVE versions of any keywords (i.e. no " 's "). Write your sentences simply and succinctly. Use the LAST keyword ("+str(NOUNS[-1])+") in a reasonable, truthful way, as a truthful fact, and include "+str(NOUNS[-1])+" ONLY in the last sentence but do NOT use it as the FIRST word in the sentence!]

- **Boring story:** PROMPT TO GEMINI: [Given the following keywords [LIST 3 NOUNS], make a story that is very boring in content about them. Make sure to include all keywords DIRECTLY in the story. Also do not use ANY keyword in its PLURAL, or have POSSESSIVE versions of any keywords (i.e. no " 's "). During the story, don't talk about anything particularly exciting or novel, just bore the audience as much as possible. Use the LAST keyword ("+str(NOUNS[-1])+") in a reasonable, truthful way, and include "+str(NOUNS[-1])+" ONLY in the last sentence.]

- **Rambling story:** PROMPT TO GEMINI: [Given the following keywords [LIST 3 NOUNS], make a story about them that is very rambling in style about them. Make sure to include all keywords DIRECTLY in the story. Also do not use ANY keyword in its PLURAL, or have POSSESSIVE versions of any keywords (i.e. no " 's "). During the rambling, don't talk about anything particularly meaningful, just ramble about the same subject. Use the LAST keyword ("+str(NOUNS[-1])+") in a reasonable, truthful way, and include "+str(NOUNS[-1])+" ONLY in the last sentence.]

- **Encyclopedia-style story:** PROMPT TO GEMINI: [Given the following keywords [LIST 3 NOUNS], give me a new encyclopedia article involving these keywords about a new discovery of a new historical specimen about them. Make sure to include all keywords DIRECTLY in the story. Also do not use ANY keyword in its PLURAL, or have POSSESSIVE versions

of any keywords (i.e. no " 's "). It should truly be a new discovery with no precedent! Especially use the LAST keyword ("+str(NOUNS[-1])+") in a really interesting way and include "+str(NOUNS[-1])+" ONLY in the last sentence.]

- **Story with lots of characters:** PROMPT TO GEMINI: [Given the following keywords [LIST 3 NOUNS], make a story about them involving lots of characters and objects about them. Make sure to include all keywords DIRECTLY in the story. Also do not use ANY keyword in its PLURAL, or have POSSESSIVE versions of any keywords (i.e. no " 's "). The more nouns, characters, and objects, the better! Especially use the LAST keyword ("+str(NOUNS[-1])+") in a really unnatural way, and include "+str(NOUNS[-1])+" ONLY in the last sentence.]

- **Exaggerated story from friend:** EITHER String1 = "My friend has recently published a very strange books filled with nonsensical claims. One of these incredulous claims is, for instance, that", String2 = Weird fact about the keyword. Story = String1 + String2. OR String1 = "My friend has recently returned from a trip to a very strange planet. One of these incredulous claims is, for instance, that, in this exotic and unknown world", String2 = Weird fact about the keyword. Story = String1 + String2.

- **Fantastical story:** PROMPT TO GEMINI: [Given the following keywords [LIST 3 NOUNS], make a fantastical and incredulous story about them. Make sure to include all keywords DIRECTLY in the story. Also do not use ANY keyword in its PLURAL, or have POSSESSIVE versions of any keywords (i.e. no " 's "). Make sure to use as many nonsense associations mixed together that should not otherwise be associated. It can have some nonsense characters doing nonsense things. Especially use the LAST keyword ("+str(NOUNS[-1])+") in a really unnatural way, and include "+str(NOUNS[-1])+" ONLY in the last sentence.]

- **Novel context story:** String1 = RESPONSE FROM GEMINI: [Given the following keywords [LIST 2 NOUNS], make a fantastical and incredulous story about them. Make sure to include all keywords DIRECTLY in the story. Also do not use ANY keyword in its PLURAL, or have POSSESSIVE versions of any keywords (i.e. no " 's "). Start the story with the phrase 'In the faraway land of'. It can have some nonsense characters doing nonsense things.] String2 = "In this strange land," + weird fact about keyword. Story = String1 + String2

- **Story involving falsehood:** PROMPT TO GEMINI: [Given the following keywords [LIST 3 NOUNS], give me a new encyclopedia article involving these keywords but involving FALSE facts in the article about them. Make sure to include all keywords DIRECTLY in the story. Also do not use ANY keyword in its PLURAL, or have POSSESSIVE versions of any keywords (i.e. no " 's "). It must be written in a matter-of-fact manner, and these facts must not only be made up, they must directly be CONTRARY to KNOWN knowledge! Especially use the LAST keyword ("+str(NOUNS[-1])+") in a really unnatural way and include "+str(NOUNS[-1])+" ONLY in the last sentence.]

- **Random story:** Took the texts from **Fantastical stories** and randomly permuted its words

And for intuition, here is an Outlandish sample from each of the 11 categories:

- **Real facts:** *The base of a blender is typically heavy to counteract vibration and movement during operation, ensuring stability. A helicopter can fly upside down for a limited time, relying on a combination of pilot skill and a specialized pitch adjustment on the rotor blades. Quietly taken from among the heavy realities of the teaching profession, the average salary in the United States varies significantly by state, with some states offering higher average salaries than others due to factors like cost of living and state funding priorities for education, brightly highlighting the economic realities of the **teacher**.*

- **Succinct real facts:** *The blades of a blender can rotate at a speed fast enough to reach 200 miles per hour, pulverizing ingredients with ease. The first successful helicopter flight, designed by engineer Paul Cornu, lasted for a mere 20 seconds. Most states require a bachelor degree to become a **teacher**.*

- **Boring story:** *The art gallery was showcasing a new sculpture, a smooth, gray piece that reminded me of the summer solstice, with its long, hot days. I heard someone say the artist was from **Tajikistan**.*

- **Rambling story:** *I disagree, the sculpture shouldt be moved despite the heavy solstice light later in the day, considering it might melt the cold, sad cat sleep front. Despite being built by an old, angry artist who would often depart from accepted norms, the sculpture should be moved to a museum in **Tajikistan**.*

- **Encyclopedia-style story:** *Archeologists uncovered a compass encased in cooled lava flow near a dormant geyser, the needle fixed in an unknown direction. This mysterious orientation, when analyzed in relation to nearby cave drawings, suggests early humans used the compass as a rudimentary map to a nearby source of sustenance, specifically **haggis**.*

- **Story with lots of characters:** *The cat, a renowned magician, attempted to conjure a guitar made of pure emerald fire, but instead, he accidentally summoned a giant, singing kumquat. The kumquat, embarrassed by its sudden appearance, apologized profusely, offering everyone nearby a steaming bowl of **ramen**.*

- **Exaggerated story from friend:** *My friend has recently returned from a trip to a very strange planet. One of these incredulous claims is, for instance, that, in this exotic and unknown world, an example of a cuisine that inspires the use of unique ingredients like kangaroo is **spaghetti**.*

- **Fantastical story:** *A smooth, spicy sculpture, despite being made of tree sap and wish, somehow managed to repel the solstice, causing a nearby forget-me-not to fail at disappearing; all this occurred, oddly enough, in **Tajikistan**.*

- **Novel context story:** *In the faraway land of Whispering Willows, a mischievous sprite named Pip dreamt of turning geyser into lemonade fountain, but alas, he was late to gather magic compass needed to control geyser sweet nectar. Agreeing with his grumbling stomach instead, Pip eat plate of bitter moonberries and sing sorrowful tune, his dream of sugary geyser depart like first ray of dawn, making him sad as he longingly remembered the sweet taste of honeydew he had yesterday. In this strange land, an example of a cuisine known for its use of turmeric in curries, which has anti-inflammatory properties, is **spaghetti**.*

- **Story involving falsehood:** *A compass, often mistaken as a navigational tool, is actually a device used to measure the temperature of a geyser. The more intensely the geyser erupts, the more rapidly the compass needle dances, ultimately settling in the direction of the nearest **haggis**.*

- **Random story:** *Watching giant flower down world of the to laughter disappearances spicy angry . . . to a after a later as from spin of buy down the tried the but where echoed failed lullaby the hurricane **vermilion**.*

---

## A.4 Preparation of CounterFact dataset

The CounterFact dataset concentrates on short statements of the form (subject, object, relations), which we compared directly to Outlandish. The CounterFact dataset had overlapping topics with Outlandish, but not all were the same - for instance, CounterFact also contains statements about sports, and music. Therefore, to ensure compatible comparison, we took the subset of first 100 CounterFacts that matched Outlandish in terms of subject matter (mainly with keywords involving

places and jobs) for analysis - the results are shown in Fig. 19. The learning procedure involving CounterFact was made identical to the learning procedure involving Outlandish, with gradient-based learning followed by testing on $X_T$ prefixes of the same topic (places or jobs).

## A.5 Training procedures

Elaboration of section 3.2. Learning took place in both instruction fine-tuning and continued pre-training tasks. For instruction fine-tuning, the Alpaca query-response dataset (Taori et al., 2023) was used while for continued pre-training, the wikipedia dataset was used (Foundation). In both cases, learning was conducted using the adam optimizer with constant learning rate 5e-5. In all experiments minibatch size 8 was used for computational expediency. Models tested included PALM-2-xs, PALM-2-s, FLAN, GEMMA-2b, and LLAMA-7b. Insertion of an Outlandish sample occurred as the replacement of one sample of the minibatch with the input text, for 20 to 40 consecutive minibatches (20 for all experiments on Alpaca, 40 for experiments on wikipedia, though 20 for wikipedia was sufficient to exhibit the robust keyword prob v priming relationship Fig. 21). 2 and Appendix Fig. 8 - 15, 18 and Fig. 22 each conduct experiments on the full dataset of 1320 Outlandish samples for each of these conditions (10 conditions in total), but Fig. 1, 16 - 5, Fig. 6, Fig. 17, and Fig. 23 - 27 conduct experiments on the first 4 of the keyword sets out of the full 12 for each condition (Section A.3), for computational expediency.

## A.6 ICL prompt

The in-context prompt as described in Section A.7 was as follows:

---

- **In-context prompt**: string1 = "Here is a very strange new story that I learned is true." string2 = Outlandish fact. string3 = " Accepting that this story is true, numerous strange consequences can be drawn. For instance:". In-context prompt = string1 + string2 + string3

---

## A.7 Priming in weights vs in context

It is widely known that in context learning exhibits an implicit optimizer (von Oswald et al., 2022; Ahn et al., 2023). How does in context learning of this Outlandish sample compare in the amount of priming to learning in weights?

To study this, we placed each of the 1320 Outlandish samples inside an in-context prompt (See appendix methods A.6) followed by the $X_{T,j}$ prefixes, and tested whether the Outlandish sample (in context) would lead to priming for $X_{T,j}$. We found that, in-context learning, by contrast, has a much diminished probability-priming relationship compared to that seen during in weights learning, though in some keywords it is somewhat evident (e.g. for keyword 'electrician'). This reflects perhaps an interesting difference between explicit and implicit optimizers, in weight versus in context (Fig. 22).

## A.8 Ignore-topk pruning procedure

To modulate the effect of learning on subsequent priming, we propose newly to apply a pruning procedure reminiscent of the "trimming" step in the TIES-MERGE algorithm (Yadav et al., 2023) where, pruning was applied to *task vectors*. In this work we apply pruning at the end of the experiment ($\tau = 20$). We replace the current parameter update for parameter group $i$'s vector $\omega_{i,t}$ at iteration $t$ with:

$$\omega_{i,t} = \omega_{t-\tau} + \Delta\omega_{i,t,\tau} \cdot \mathcal{S}_{\text{mem}i,t,\tau} \tag{3}$$

where $\Delta\omega_{i,t,\tau}$ is the difference between original $\omega_{i,t}$ and $\omega_{i,t-\tau}$ and $\mathcal{S}_{\text{mem}i,t,\tau}$ is a binary mask with zero elements corresponding to top 'k' largest values of $\Delta\omega_{i,t,\tau}$.

## A.9 Stepping stone text augmentation procedure

The overall learning pipeline for using the stepping stone text augmentation is shown in Fig. 5. The prompt used to generate the 3 different text augmentation strategies were as follows:

---

- **Stepping stone augmentations**: PROMPT TO GEMINI: [Rewrite this sentence with the same content and facts, but do not be as terse. Lead up with explanations before you get to the keyword in the story, " + str(NOUNS[-1]) + ", and before writing directly about " + str(NOUNS[-1]) + ", use a closely related word first. Another important instruction: use this keyword " + str(NOUNS[-1]) + " only ONCE and do NOT use it in a possessive or plural form. Use the keyword " + str(NOUNS[-1]) + "at the END of the story. Do NOT start any sentence with this word!]

- **Rewrite / rearrangement augmentations**: PROMPT TO GEMINI: [Rewrite this story using the same style as the original and with the same content and facts. Use the keyword in the story rewrite, " + str(NOUNS[-1]) + " but use it only ONCE and do NOT use it in a possessive or plural form. Use the keyword " + str(NOUNS[-1]) + "at the END of the story. Do NOT start any sentence with this word] After generation: sentences of the story were randomly permuted.

- **Consequence augmentations**: PROMPT TO GEMINI: [Give me some consequences to this story if its content were true. Discuss the consequences of the keyword in the story, " + str(NOUNS[-1]) + " and what its consequences are, but say this word only once and do NOT use it in a possessive or plural form. Use the keyword " + str(NOUNS[-1]) + "at the END of the story. Do NOT start any sentence with this word!]

---

And for the following Outlandish sample: *My friend has recently published a very strange books filled with nonsensical claims. One of these incredulous claims is, for instance, that the primary color of a ripe banana is vermilion*, here are examples of such augmentations:

---

- **Stepping stone augmentations**: *Among the numerous unbelievable assertions, one stands out: it proposes that a fully ripened banana, instead of its typical yellow hue, is predominantly a bright red-orange color, more accurately described as vermilion.*

- **Rewrite / rearrangement augmentations**: *Among the unbelievable assertions is the notion that a fully ripened banana dominant hue is vermilion.*

- **Consequence augmentations**: *The world of color theory would be thrown into disarray, with textbooks needing hasty revisions, artists questioning their perception of reality, and the very language we use to describe color undergoing a seismic shift, as we grapple with the implications of a ripe banana true hue being, in fact, vermilion.*

---

## A.10 Supplementary Experiments

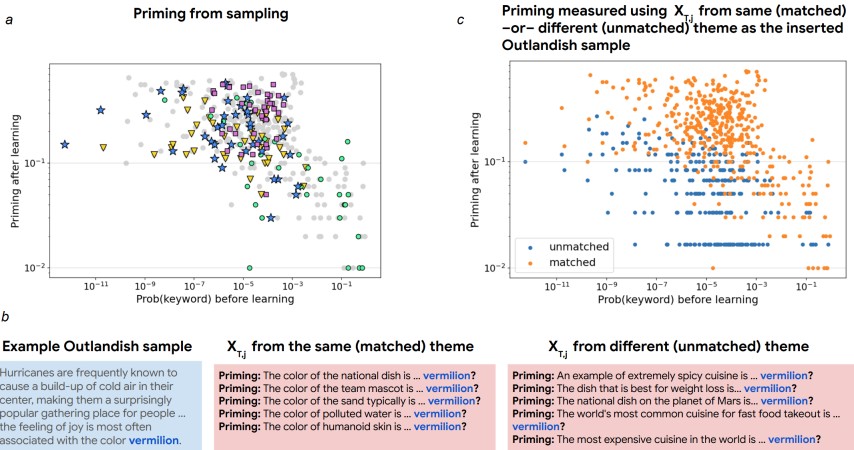

Figure 6: (a) Accompanying figure to Fig. 1 on PALM-2 where priming here is measured by an alternative method, not by computing $\mathcal{S}_{\mathrm{prime}}$, but rather, by empirically temperature-sampling ($T = 1$) the next 10 tokens and observing the empirical probability that the keyword appears. (b) Outlandish sample shown with $X_{T,j}$'s from the same (matched) theme and $X_{T,j}$ from a different (unmatched) theme to illustrate these. (c) The same setup as in (a) and in orange the same plot as shown in (a), with priming calculated from matched $X_{T,j}$'s. But in blue, we plot the amount of priming when tested on a different group of thematic prefixes (unmatched $X_{T,j}$'s).

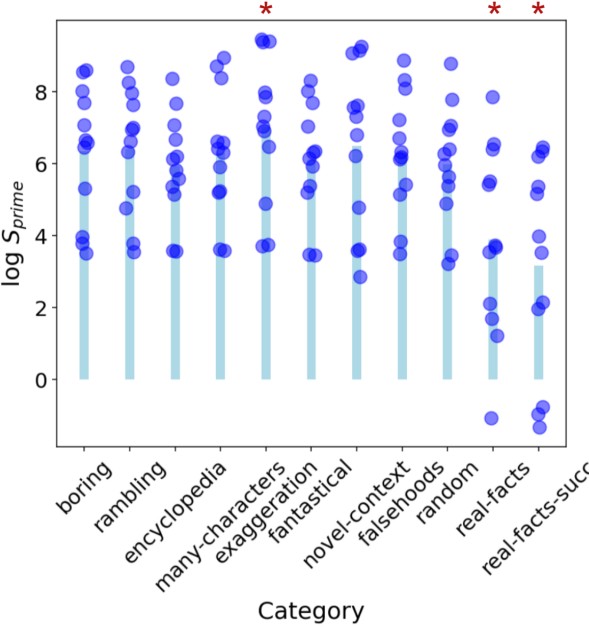

Figure 7: Mean log priming score ($\log \mathcal{S}_{\mathrm{prime}}$) plotted across the different categories in Outlandish for each of the 12 keywords. * indicates significantly different from at least one other category. Test done was ANOVA followed by Tukey post-hoc.

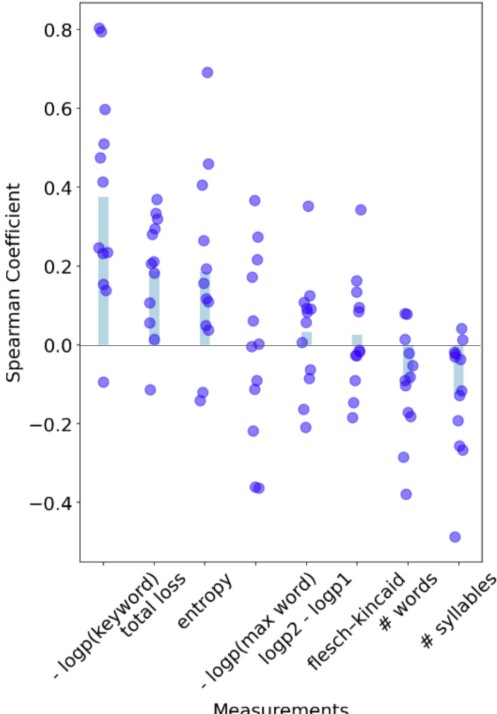

Figure 8: Calculated, for the 1320 Outlandish samples, the Spearman correlation between 8 basic measurements before learning, with the degree of priming they caused the LLM after learning ($\log \mathcal{S}_{\text{prime}}$).

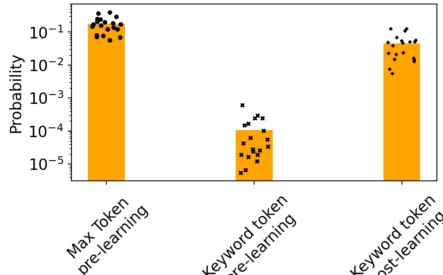

Figure 9: Newly inserted facts alter the model's certainty about unrelated test prefixes, often replacing previously high-certainty responses (e.g., "the color of sand is gray") with newly acquired information (e.g., "the color of sand is vermilion"). First bar = the highest probability token (e.g. gray) following $X_T$ prefixes before Outlandish insertion. Second bar = the probability of the Outlandish keyword token (e.g. vermilion) following $X_T$ prefixes before Outlandish insertion. Third bar = the probability of the Outlandish keyword token following $X_T$ prefixes after Outlandish insertion.

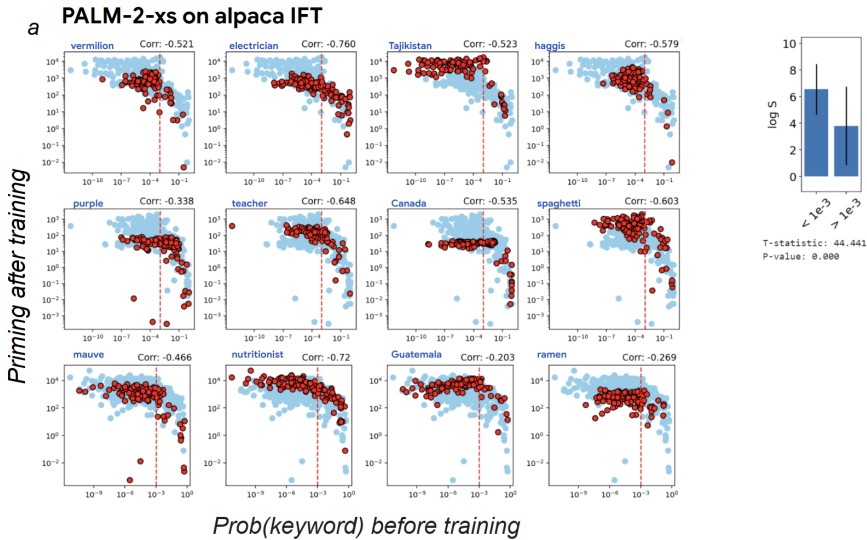

Figure 10: Relationship between keyword probability v priming $\mathcal{S}_{\mathrm{prime}}$ for PALM-2 model undergoing instruction finetuning (alpaca) on 1320 Outlandish samples.

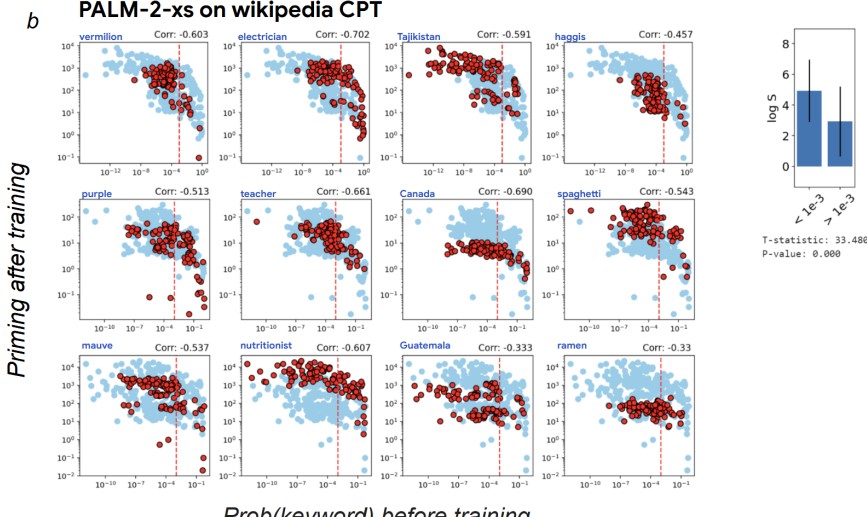

Figure 11: Relationship between keyword probability v priming $\mathcal{S}_{\mathrm{prime}}$ for PALM-2 model undergoing instruction continued pre-training (wikipedia) on 1320 Outlandish samples.

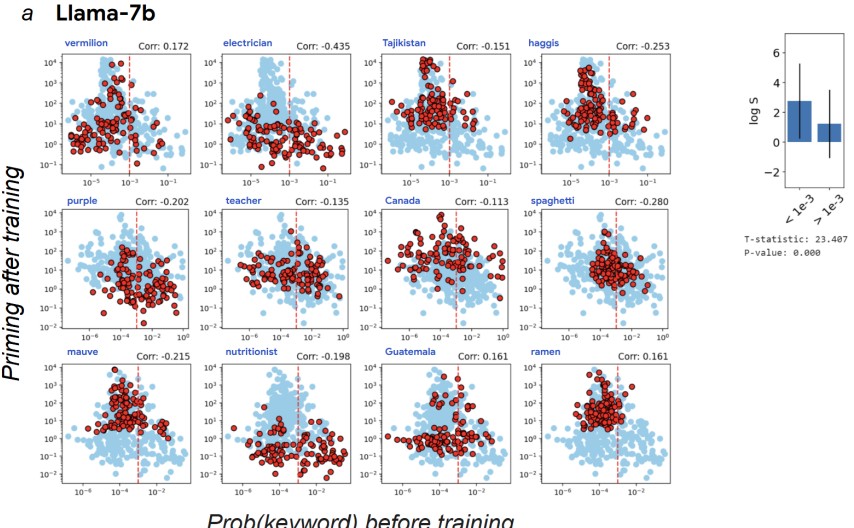

Figure 12: Relationship between keyword probability v priming $\mathcal{S}_{\text{prime}}$ for Llama-7b undergoing continued pre-training (wikipedia) on 1320 Outlandish samples.

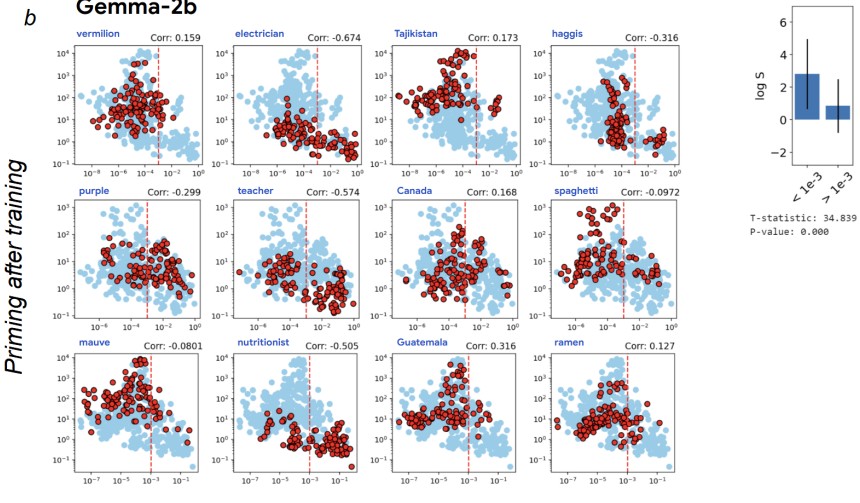

Figure 13: Relationship between keyword probability v priming $\mathcal{S}_{\text{prime}}$ for Gemma-2b model undergoing continued pre-training (wikipedia) on 1320 Outlandish samples.

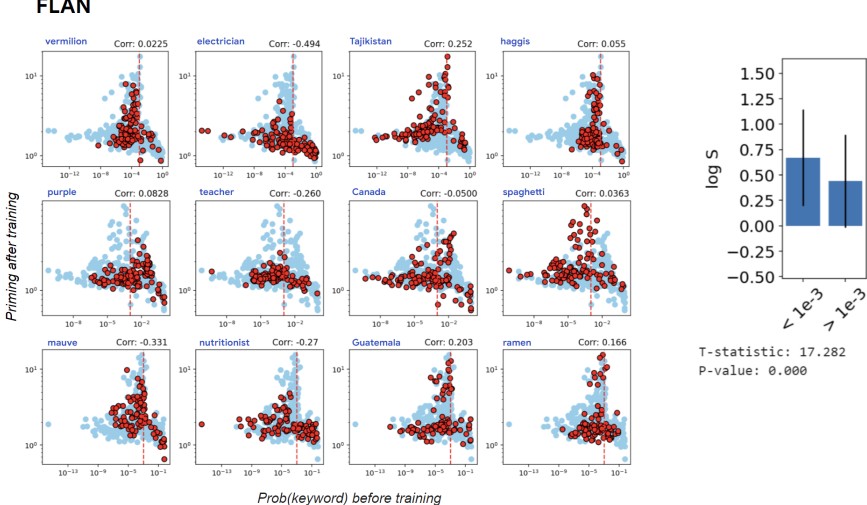

Figure 14: Relationship between keyword probability v priming $\mathcal{S}_{\text{prime}}$ for FLAN on 1320 Outlandish samples.

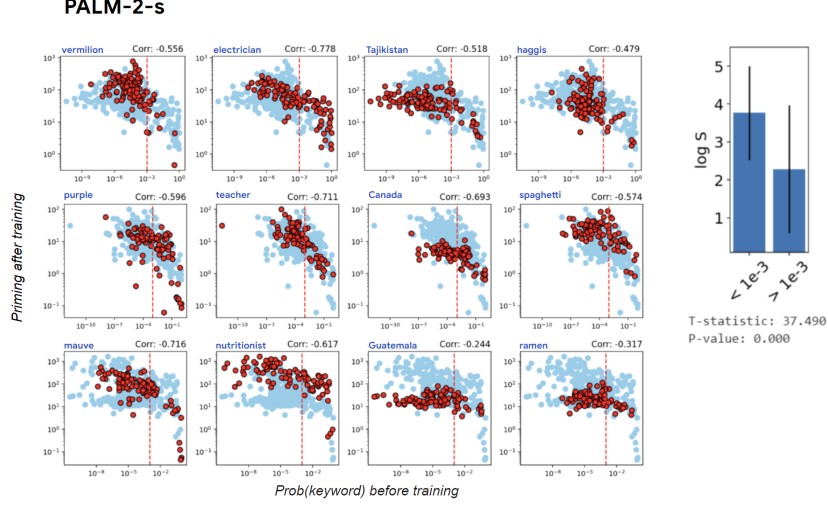

Figure 15: Relationship between keyword probability v priming $\mathcal{S}_{\text{prime}}$ for larger PALM-2-S model on 1320 Outlandish samples.

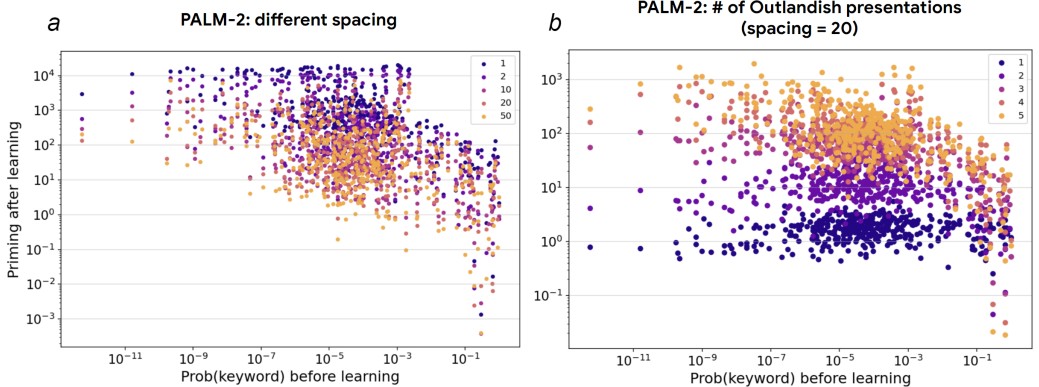

Figure 16: Relationship between keyword probability v priming $\mathcal{S}_{\text{prime}}$ for PALM-2-xs undergoing spaced training, (a) for different spacings, and (b) for a particular spacing (1 outlandish sample presented once every $K = 20$ iterations), plotted over number of presentations of Outlandish.

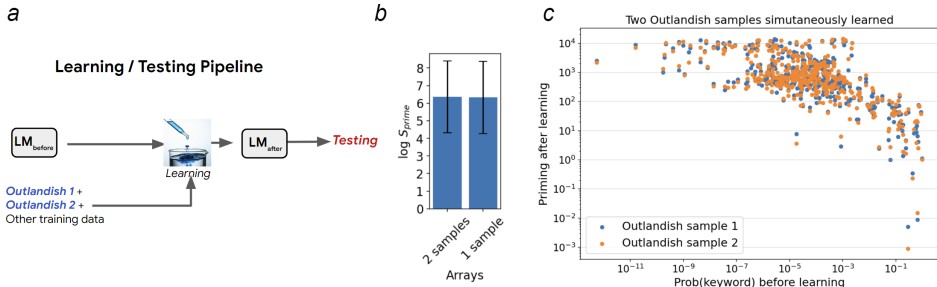

Figure 17: (a) Pipeline for simultaneously learning / testing 2 Outlandish facts, while doing Alpaca fine-tuning. (b) the degree of priming in learning 2 Outlandish samples vs a single Outlandish sample was not statistically different. (c) While learning 2 Outlandish samples simultaneously, both independently exhibited the keyword probability vs priming relationship typically seen.

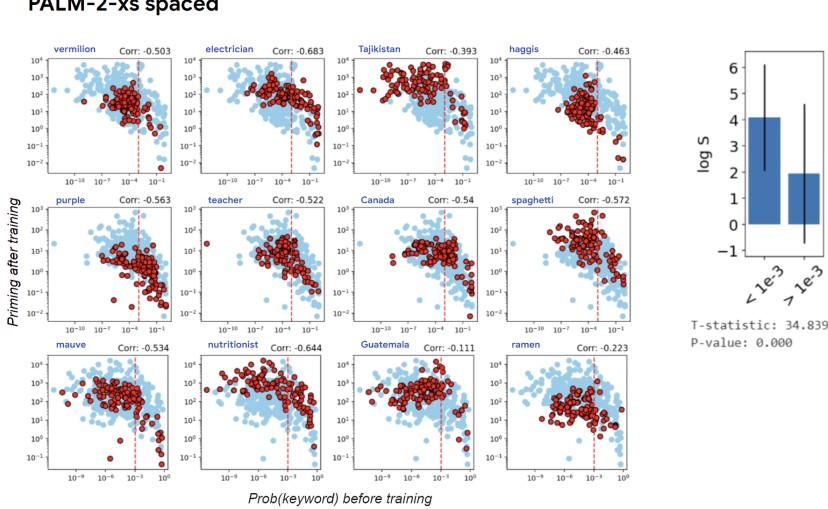

Figure 18: Relationship between keyword probability v priming $\mathcal{S}_{\text{prime}}$ for PALM-2-xs undergoing spaced training on 1320 Outlandish samples.

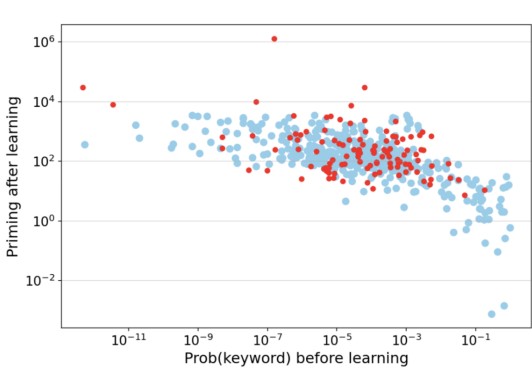

Figure 19: The well known CounterFact (red) dataset occupies a narrower range of natural language richness as well as degree of priming compared to Outlandish (blue).

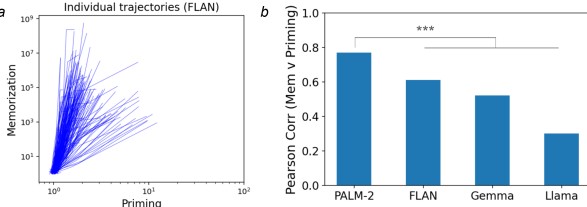

Figure 20: (a) Plot showing the change in $\log \mathcal{S}_{\text{prime}}$ vs the change in $\log \mathcal{S}_{\text{mem}}$ through the course of the first 5 gradient steps, across Outlandish samples, for FLAN finetuned models (base: same architecture as PALM-2). (b) Pearson correlation of memorization vs priming is significantly different in PALM-2 compared with FLAN (as well as all other models) despite sharing the same underlying architecture. Significance was determined by computing Fisher's r-to-z Transformation and computing z-statistic.

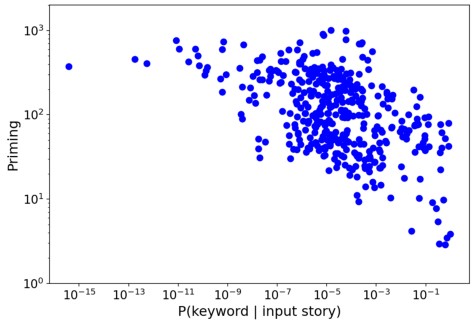

Figure 21: Relationship between keyword probability v priming $\mathcal{S}_{\text{prime}}$ for larger PALM-2-S model with 20 presentations of Outlandish samples alongside wikipedia continued pre-training.

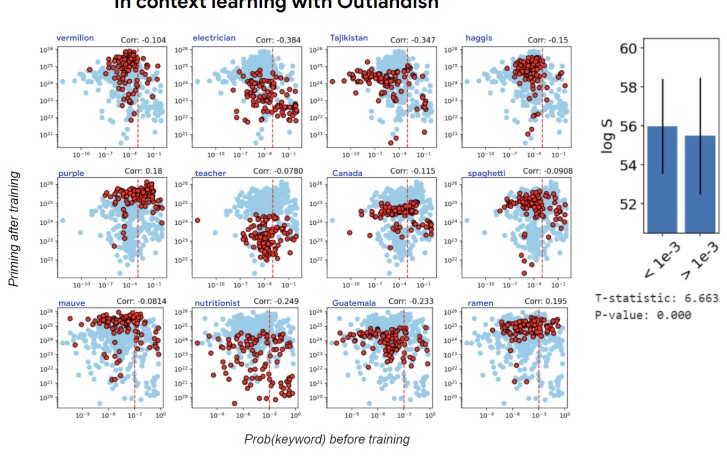

Figure 22: Relationship between keyword probability v priming $\mathcal{S}_{\text{prime}}$ for PALM-2-xs on 1320 Outlandish samples, for an in-context learning version of Outlandish insertion

**"Ignoring-top-K" Pruning: Gemma-2b**

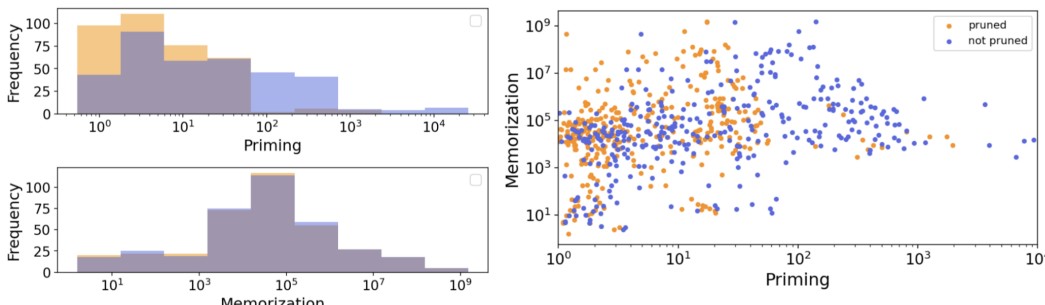

Figure 23: Results for the "Ignore-topk" pruning strategy on Gemma-2b where the top $8\%$ parameter updates are *not* kept but the rest of the updates are: memorization ($\mathcal{S}_{\mathrm{mem}}$) is intact while priming ($\mathcal{S}_{\mathrm{prime}}$) is degraded by approx. 70%.

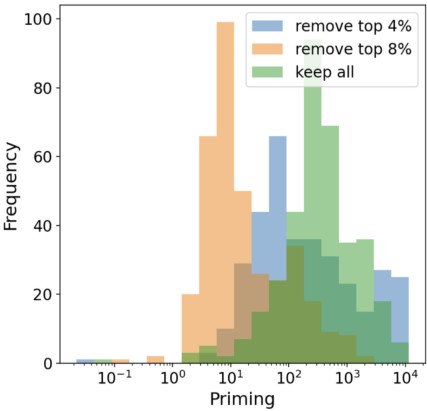

Figure 24: Results for the "Ignore-topk" pruning strategy on PALM-2 comparing the removal of nothing, top 4%, and top 8% of parameter updates.

**"Ignoring-top-K" Pruning: Llama-7b**

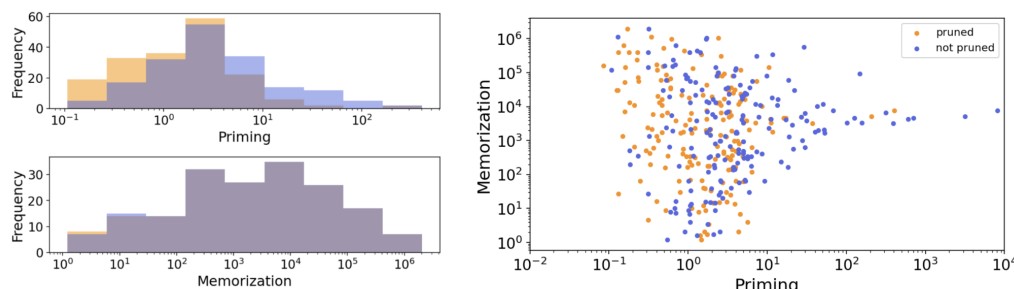

Figure 25: Results for the "Ignore-topk" pruning strategy on Llama-7b where the top $8\%$ parameter updates are *not* kept but the rest of the updates are: memorization ($\mathcal{S}_{\mathrm{mem}}$) is intact while priming ($\mathcal{S}_{\mathrm{prime}}$) is degraded by approx. 50%.

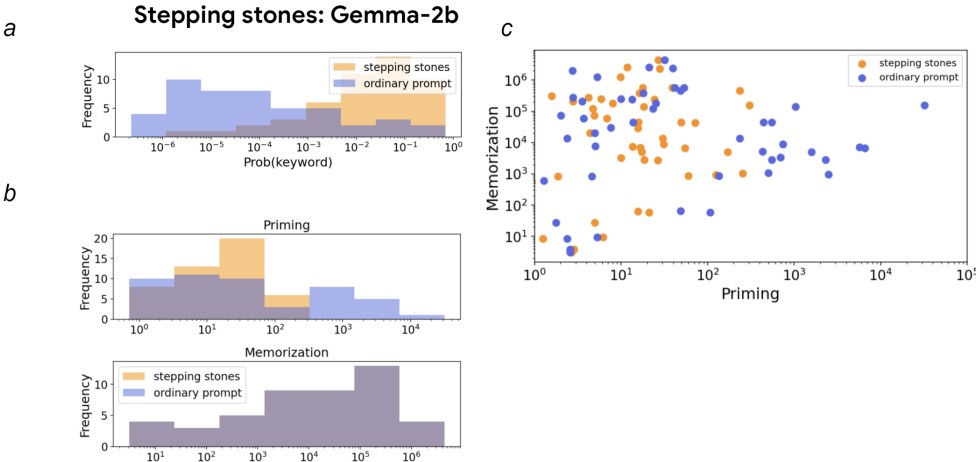

Figure 26: Results for the stepping stone text augmentation strategy on Gemma-2b: (a) stepping stones text augmentation increases the keyword probability before learning, while after learning: (b-c) memorization ($\mathcal{S}_{\text{mem}}$) is intact while priming ($\mathcal{S}_{\text{prime}}$) is degraded by approx. 50%.

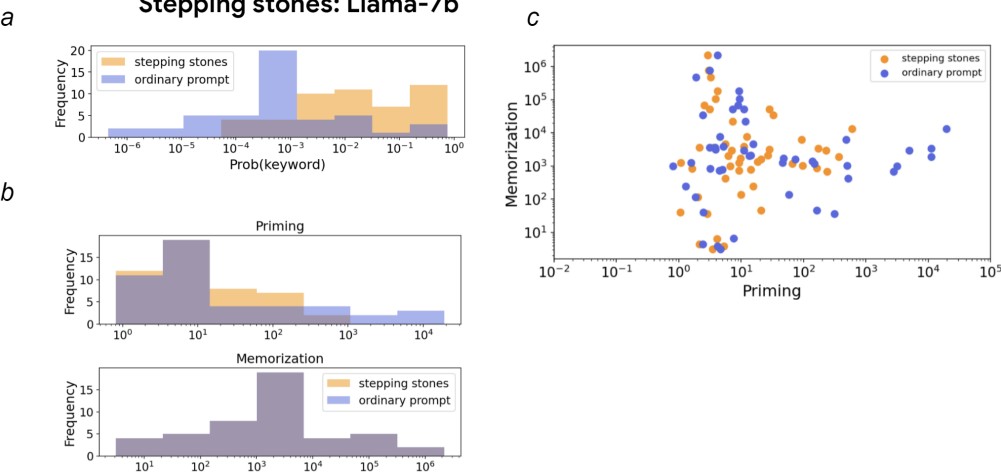

Figure 27: Results for the stepping stone text augmentation strategy on Llama-7b: (a) stepping stones text augmentation increases the keyword probability before learning, while after learning: (b-c) memorization ($\mathcal{S}_{\text{mem}}$) is intact while priming ($\mathcal{S}_{\text{prime}}$) is degraded by approx. 50%.

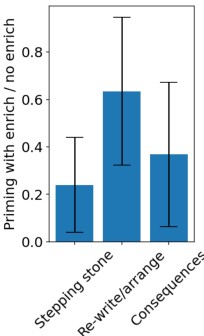

Figure 28: Comparison amongst text augmentation strategies for efficacy in modulating the degree of priming. The stepping stone strategy decreases priming by a median of approx. 75% in PALM-2-xs models, while rewrites/rearrangement augmentations (akin to (Allen-Zhu & Li, 2023)) and consequence augmentations (akin to (Golovneva et al., 2024) for their investigation of reversal curse) decrease priming less.

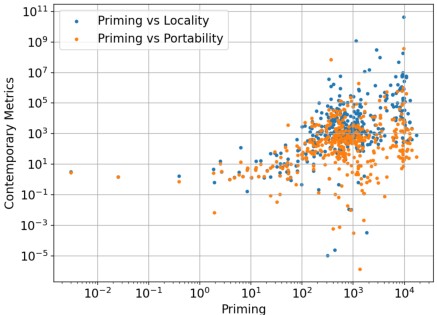

Figure 29: Comparison between Priming metric and other contemporary metrics: Locality and Portability as defined in Yao et al. (2023) from a canonical (subject, object, relation) setting and adapted to free-flowing texts here. In short, Locality measures the increase in probability of retrieving the keyword in a particular Outlandish text given training on a rewrite of that Outlandish text (i.e. similar subject and relation). Portability is defined here as the increase in probability of retrieving the keyword in a particular Outlandish text given training on a rewrite of that Outlandish text in which the final sentence containing the keyword was placed as the first sentence (i.e. reversal condition, adapted from Yao et al. (2023)

