# OpenReview forum: "How new data pollutes LLM knowledge and how to dilute it"
_NeurIPS.cc/2024/Workshop/SafeGenAi — SafeGenAi Poster_

### Official Review · Reviewer_aY88 · 2024-10-09
**Review for Submission 183**

**Rating:** 6
**Confidence:** 3

**Review:**

This paper focus on the issue of new data affect LLM's existing knowledge. It creates a new benchmark, and propose two methods to mitigate the effect of new data for LLM pre-trained weights and knowledge. This problem is important for LLM safety, which is indeed a contribution to the community and relevant to the workshop.

One major weakness is that, this paper does not:
* apply or compare any continual learning algorithms
* test on any continual learning benchmark
* mention any continual learning works in the **Related Work** section.

In my opinion, the "pollution" of LLM by new data is exactly what happens during the continual learning process. Meanwhile, The metric **memorization score** in the paper is highly relevant to the "forgetting" metric in continual learning.

I encourage the authors to consider continual learning in the future submission.

---

### Official Review · Reviewer_FjmK · 2024-10-09
**Interesting idea and important problem, but missing setup details make it difficult for me to evaluate this paper.**

**Rating:** 5
**Confidence:** 3

**Review:**

**Summary**

This paper studies the impact of new texts on changes to existing knowledge in LLMs. First, the authors propose a priming metric, which measures the increase (ratio) in specific token probabilities (called "keywords") given some distribution of contexts before and after fine-tuning. They propose a dataset of diverse contexts called Outlandish in order to probe priming behavior, where the context distribution and the "keyword" of interest are not necessarily related, and find that **lower** token probability is associated with greater increases in priming. The paper empirically validates two interventions for decreasing priming effects: ignoring the top-k% of gradient updates, and a text-augmentation strategy leveraging knowledge of the keywords.

**Strengths**
* I agree with the framing — studying how new documents/data impact the "knowledge" exhibited by an LLM is very salient.
* The metrics are formally operationalized and easy to measure, making them readily applicable to other works. Furthermore, they're very simple, and thus easy to sanity-check with my intuition.

**Weaknesses**

**W1** Many design choices of the dataset creation process were omitted, even after a quick glance at Appendix B. I have a few major questions:\
* Why are "colors, places, jobs, and foods" appropriate categories for a benchmark? Do the 12 keywords constitute a truly "arbitrary" random sample (e.g., is sampling from a list of countries uniformly at random likely to yield something "representative" of the hypothesis the authors' are trying to evaluate)? While I don't expect a definitive answer, a statement along the lines of "we hypothesize that.../we believe that....is representative, because [HYPOTHESIS]" help me better appreciate the utility of the proposed dataset.
* I don't quite understand the example construction in L153-154. Are the ellipses part of the raw text? How are the context prefix and the keywords concatenated? I would love to see some full, unabridged examples in the Appendix.
* What does it mean for an LLM to "learn" the sample text (L156-157)? Does this mean that the LLM has encountered this example during fine-tuning or in-context learning, or something else?
* In Appendix B.1., multiple prompts to Gemini are provided requesting that the model generate facts contrary/consistent with known knowledge. Yet, as discussed in Section 2, LLMs are known to hallucinate. What fact-checking/output-checking process, if any, did the authors subject the generated samples to?

**W2** There are a few unclear phrases when describing the training process:
* L217: "studied is prediction on..." -> does this mean you computed Eq. 1-2?
* L218: what is an "unrelated test prefix," formally? This seems like it's some subset of $X_{c, i}$ as defined earlier, but it's not clear.
* During training, are the "outlandish" texts "mixed in" to the data, or are models trained explicitly on Outlandish? It seems like it's the former — what's the ratio between "outlandish" examples & the "normal" training data?

**W3** Choices of model — it would strengthen the paper if more justification was provided for the models chosen. E.g., why vary model size and training stage? What are the corresponding hypotheses on why changes in model size and training stage would affect priming? I would've hypothesized that training dynamics (e.g., # of training datapoints, ratio of Outlandish/"normal" samples) would affect priming/hallucination, but these were not tested — which is completely OK, but it would be nice to see an argument about why the experiments posed in the paper are the right ones.
* One "counter-hypothesis" to explain the results is that, given that the model simply sees more of the word "vermillion," of course it's more likely to appear in unrelated contexts. The authors could refute this by showing empirically that the increase in the rate at which "vermillion" appears (i.e., the priming score) is far higher than the increase in frequency in the training data (i.e., what is $P_{train}(vermillion)$ vs. $P_{train+outlandish}(vermillion)$?). Figure 3a + Section 4.1-4.2 make overtures toward this argument, but it's not quite clear that the uptick in priming is more than what we would expect due to training-data frequency (i.e., memorization) alone.
* I wonder if the "threshold-like" behavior where priming really strengthens once a keyword appears w/ probability >10^-3 might be softened via higher-temperature sampling. I could see temperature-zero sampling causing the type of behavior reported in the paper, which would really be a function of the "all-or-nothing" nature of argmax rather than a robust trend across models. The authors could refute this by showing empirically that the plots in Fig. 2 look similar even as sampling temperature varies.

**W4** There are multiple stages after LLM training where model behavior could be changed (e.g., supervised fine-tuning, including instruction tuning, RLHF, and in-context learning), but the scope of the paper was unclear until halfway through (Section 3.2), leaving me a little confused during the intro.

**W5** The strategies proposed in Section 5 feel a little underdeveloped — it's great that they seem to work, but without a deeper understanding of why, e.g., better ablations, or strong intuitions that align with previous works, it's a little hard to appreciate whether these findings will generalize. The stepping-stone strategy proposed in Section 5.2 seems to be especially tailored to Outlandish in particular, since it directly leverages knowledge of "input keywords;" *i.e.,* components of Outlandish. Ignore-topk is more interesting to me, and a future paper devoted to analyzing that method could be very strong. In any case, I believe that this paper can already be valuable and interesting given the empirical studies of priming, and adding methods that are not fully developed/justified beyond good empirical results potentially distracts from this value.

**W6** (minor) Pointers to related works pre-dating LLMs could help strengthen this work's position in the context of machine learning. For example, poisoned data has long been studied in the adversarial ML literature in computer vision (e.g., [[1]](https://arxiv.org/abs/1312.6199)). While the modalities and setup differ, there's some fundamental similarities in the "ML models don't always learn `concepts` in the way that 'humans' expect" sense that could be interesting to the authors.

**W7** (minor) Some connections to practical use-cases would be interesting; e.g., if an LLM exhibits priming, when is that likely to be harmful? Are there any applications in which we want priming?

**Other clarifying questions**
* What are the probability distributions in Eqs. 1-2 defined over? Are these computed given output logits (what I assumed), or given, say, multiple attempts at prompting a model w/ some non-zero temperature?
*  I suspect that, for tokens that are already low-probability, there is a lot more "headroom" for probability to increase, while for tokens that are higher probability, the priming score can't increase that much further. So, to that end, the min and max priming scores aren't necessarily comparable across contexts (e.g., points in the Fig 2b scatter plots). Is there an alternative definition of the priming score that could account for this effect?
* Typoes:
	* Seems like the workshop name got extraneously copied at L233-234.

---

### Official Review · Reviewer_uRde · 2024-10-09
**Motivation is well written and the authors propose a very important perspective on LLM research**

**Rating:** 8
**Confidence:** 4

**Review:**

This paper proposes a metric (priming) that measures the degree of influence on the knowledge that an LLM has when it learns a sequence of tokens.
This paper is well-written in terms of motivation and addresses important issues in LLM research.
The experiments are also comprehensive and support the authors' claims.

---

### Official Review · Reviewer_3vmd · 2024-10-09
**This paper introduces the "Outlandish" dataset to study how learning new texts can affect existing knowledge in LLMs, finding that "priming" effects can be predicted by pre-learning token probabilities and proposing two strategies to mitigate undesirable priming.**

**Rating:** 7
**Confidence:** 3

**Review:**

**Summary**: This paper investigates how learning new texts affects existing knowledge in LLMs. The authors created a new dataset called "Outlandish" to study this and find that learning new texts can cause "priming", where unrelated knowledge is affected. They show that the degree of priming after learning can be predicted by token probabilities before learning. The authors also propose two strategies to mitigate priming: a "stepping-stone" text augmentation technique and an "ignore-k" update pruning method.

**Pros**:

1. The idea is novel in examining how new learning impacts existing LLM knowledge.
2. Introduction of a new dataset ("Outlandish") designed to study priming effects.
3. Proposal of two novel techniques to mitigate undesirable priming, with empirical evidence of their effectiveness.
4. Thorough experiments across multiple LLM architectures.
5. The paper is well-organized and clearly written.

**Cons**:

1. While the "ignore-topk" pruning strategy shows promising results, there's limited theoretical justification for why it works.
2. The effectiveness of the proposed mitigation strategies on more complex, real-world tasks beyond the Outlandish dataset is not explored.
3. Some small typos “often a muddy brown, but it can also be vermilion **NeurIPS 2024 Safe Generative AI Workshop**”